# Chitosan-Coated Bacterial Cellulose (BC)/Hydrolyzed Collagen Films and Their Ascorbic Acid Loading/Releasing Performance: A Utilization of BC Waste from Kombucha Tea Fermentation

**DOI:** 10.3390/polym14214544

**Published:** 2022-10-26

**Authors:** Pantitra Yakaew, Thapani Phetchara, Piyaporn Kampeerapappun, Kawee Srikulkit

**Affiliations:** 1Petrochemistry and Polymer Science, Faculty of Science, Chulalongkorn University, Bangkok 10330, Thailand; 2Department of Materials Science, Faculty of Science, Chulalongkorn University, Bangkok 10330, Thailand; 3Faculty of Textile Industries, Rajamangala University of Technology Krungthep, Bangkok 10120, Thailand; 4Center of Excellence on Petrochemical and Materials Technology, Chulalongkorn University, Bangkok 10330, Thailand

**Keywords:** SCOBY bacterial cellulose/hydrolyzed collagen film, chitosan coating, mechanical properties, antibacterial activity, ascorbic acid loading and release

## Abstract

SCOBY bacterial cellulose (BC) is a biological macromolecule (considered as a by-product) that grows at the liquid–air interface during kombucha tea fermentation. In this study, BC:HC (hydrolyzed collagen) blend films coated with 1 wt% chitosan (CS) were loaded with ascorbic acid to study loading/releasing performance. At first, the mechanical properties of the blend films were found to be dependent on HC ratio. After chitosan coating, the coated films were stronger due to intermolecular hydrogen bonding interaction and the miscibility of two matrixes at the interface. The antibacterial activity test according to the AATCC Test Method revealed that chitosan-coated BC/HC films exhibited excellent antimicrobial activity against *S.aureus* growth from the underneath and the above film when compared to BC and BC:HC films. Moreover, chitosan was attractive to ascorbic acid during drug loading. Consequently, its releasing performance was very poor. For BC:HC blend films, ascorbic acid loading/releasing performance was balanced by water swellability, which was controlled using blending formulation and coating. Another advantage of BC films and BC:HC blend films was that they were able to maintain active ascorbic acid for a long period of time, probably due to the presence of plenty of BC hemiacetal reducing ends (protective group).

## 1. Introduction

Bacterial cellulose (BC), one type of interpenetrating polymer composed of 100% cellulose, is widely studied in terms of culturing and in applications such as biomedical [1,2,3,4,5,6], composites [7,8,9,10,11,12,13,14,15], environmental remediation [16,17,18,19,20,21], tissue engineering [22,23,24], pharmaceutical [25,26,27,28], packaging [29,30,31], textiles [32,33,34], and catalysts [35]. Due to a networked nanofibrous structure, BC can be fabricated into various forms including hydrogels, aerogels, and films. When focused on the BC structure, the three-dimensional network is derived from physically intermolecular hydrogen bonding, which is chemical crosslink-free (non-toxic hydrogel). Advantageously, BC exhibits high water absorption capability and liquid retention when compared to its dry weight in a similar manner to chemically cross-linked hydrogel. Symbiotic culture of bacteria and yeast (SCOBY) BC is a cellulosic by-product that is considered as waste from kombucha tea (which is a fermented drink from tea, sugar, bacteria, and yeast). Compared to *Nata de coco,* which is a food hydrogel, SCOBY BC is a bacterial cellulose hydrogel formed on the air–culture medium liquid interface; the longer the fermentation time, the thicker the floating BC hydrogel. In contrast to *Nata de coco*, the SCOBY BC is either discarded as waste or collected to be employed as a pellicle starter for the next kombucha tea fermentation. It was found that SCOBY BC exhibited high crystallinity, excellent film formation, and a matrix for bio-composites [9,10]. SCOBY BC can be molded into various sizes and forms depending on the culture container. Since BC hydrogel has the ability to absorb and retain liquid about 40–50 times its own weight [1], it has gained much attention in the fields of biomedical, tissue engineering, and pharmaceutical applications. The advantage of BC dressing compared to gauzes is that it is non-adherent and easy to peel off without pain. However, BC is not considered as an active wound dressing due to its lack of antimicrobial activity and bioactive performance, which are found in other bioactive materials such as chitosan/its derivatives, collagen (commonly found in skin care products), et al. [36,37,38]. Chitosan is a natural cationic polymer that is well-known for its antimicrobial activity against Gram-positive bacteria such as *S. aureus* commonly found in infected wounds.

Therefore, in this study, SCOBY BC hydrogel was collected from kombucha tea fermentation waste. The obtained SCOBY BC/hydrolyzed collagen (HC) films were prepared by mechanical blending and casting. Note that collagen presents in a non-soluble form that is immiscible with BC; hence, HC was suitable in this study. Finally, dried films were coated with 1% (*w/v*) acidic chitosan solution, resulting in flexible and resilient films. The mechanical and physical properties of resultant films were evaluated. The antimicrobial activity against *S. aureus* and ascorbic acid loading/releasing performance were investigated, aiming at active wound dressing application.

## 2. Materials and Methods

### 2.1. Materials and Reagents

Peptone was purchased from Sisco Research Laboratories Ltd., Maharashtra, India. Yeast extract powder was purchased from HiMedia Laboratories Pvt. Ltd., Maharashtra, India. Kombucha SCOBY pellicle starter was purchased from local kombucha tea fermentation makers. Chitosan flake was purchased from Bonafides Marketing Co., Ltd., Bangkok, Thailand. Hydrolyzed collagen (HC) was bought from Meiji Co., Ltd. (Tokyo, Japan).

### 2.2. Static Culture of SCOBY BC Hydrogel 

A static culture of SCOBY BC hydrogel was carried out according to our previous report [9]. Prior to BC incubation, a plastic container mold (20” (L) × 15” (W) × 12” (H)) was cleaned and sun dried. Culture medium (2 L) was composed of 5 g/L peptone, 5 g/L yeast extract powder, and 50 g/L sugar. The culture liquid was sterilized at boiling temperature for 30 min and then allowed to cool down to room temperature. After that, kombucha SCOBY pellicle and starter liquid (250 g) and raw vinegar (200 mL) were poured into the culture medium container. The container was covered with woven cotton fabric and left to stand in open air without direct sunlight for about 15 days to allow BC hydrogel to form on the air–liquid interface, as shown in Figure 1. BC hydrogel was cleaned by boiling in diluted NaOH and bleached using hypochlorite to remove impurities such as by-products, bacteria, and colorant. The obtained white BC hydrogel was kept for the preparation of BC/HC films in the next step.

### 2.3. Preparation of Chitosan Coated BC/Hydrolyzed Collagen Films

SCOBY BC/HC films at the BC:HC wt% ratios of 90:10, 80:20, and 70:30 in the presence of 5 wt% glycerol as a plasticizer were prepared by mechanical blending and casting. An example of BC90:HC10 was prepared as follows: 1.98 g of BC (based on dry weight) and 0.30 g of HC was physically mixed. Then, 150 mL of deionized water was added, and the mixture was homogenously blended using a high-speed homogenizer. After that, the dispersion was centrifuged to remove air bubbles. The supernatant liquid was discarded. The collected viscous paste was thinned by the addition of deionized water prior to casting onto the plastic mold (9” (L) × 8.5” (W)) and left to dry freely. Finally, the evenly smooth BC/HC film was coated with 20 mL of 1% (*w/v*) acidic chitosan solution and allowed to dry freely in open air.

### 2.4. Characterizations

The functional groups of BC, BC/HC films, and BC/HC films coated with chitosan were analyzed by Fourier transform infrared spectroscopy using a transmission mode (Nicolet 6700 FT-IR, Thermo Fisher Scientific, Waltham, MA, USA) recording wave-number of 4000–400 cm^−1^ with a resolution of 4 cm^−1^. Note that a sample was physically mixed with KBr powder and compressed into a transparent KBr disk prior to characterization. Scanning electron microscopy (SEM) was performed using a scanning electron microscope (SEM JEOL model JSM-6400LV, Tokyo, Japan) at an accelerating voltage of 15 kV. Samples were freeze-dried using Christ model beta 1–8 LD plus and kept dried prior to characterization. 

Water absorption was carried out by the immersion method. The triplicate samples before immersion (dry) and after immersion were weighed and subsequently calculated and averaged as in Equation (1): percent water uptake (%) = (W_af_ − W_bf_)/W_bf_ × 100(1)

### 2.5. Properties

#### 2.5.1. Mechanical Properties

Mechanical properties of the films were evaluated by a universal testing machine (5ST model, Tinius Olsen Ltd., Redhill, UK) according to the ASTM D882. The films with a rectangular shape (100 × 15 mm) were stretched with a gauge length of 50 mm, load cell of 500 N, and crosshead speed of 10 mm/min. 

#### 2.5.2. Antimicrobial Activity 

Antimicrobial activity was evaluated according to the AATCC TM 147 standard method.

#### 2.5.3. Loading Capacity of Coated Films

To determine the ascorbic acid loading efficiency, 250 mg of ascorbic acid was mixed with 50 mL of phosphate buffered saline (PBS, pH 7.4). The films with a test area of 1 cm^2^ were soaked in 5 mL of ascorbic acid solution for 12 h at 37 °C. After that, the films were blotted with filter paper to remove excess water and measured by an UV-Vis spectrophotometer (Labtech, Hopkinton, MA, USA) at λ_max_ = 265 nm. The ascorbic acid loading efficiency, amount of adsorbate, and loading capacity were calculated using the following formula [39]:(2)$$Ascorbic\;acid\;loading\;efficiency\;(\%)=\frac{A_{0}-A}{A}\times 100$$
where *A_0_* is the absorbance of the initial solution and *A* is the absorbance of the tested solution.
(3)$$Amount\;of\;adsorbate=\frac{Ascorbic\;acid\;loading\;efficiency\;\left(\%\right)}{100}\times 25\;\mathrm{mg}$$
(4)$$Loading\;capacity=\frac{Amount\;of\;adsorbate\;\left(\mathrm{mg}\right)}{Amount\;of\;adsorbent\;\left(g\right)}$$

#### 2.5.4. Drug Release Tests

A coated film was immersed in 15 mL of PBS solution (pH 7.4) and shaken with a frequency of 100 rpm at 37 °C. In total, 3 mL of sampling was removed and stored as stock to measure the absorption at 1 h intervals for 6 h. After sampling, the equal volume of fresh buffer was added to the solution to maintain the original volume. The absorbance of each sampling solution was measured at λ_max_@265 nm using a UV-Vis spectrophotometer. The amount of released ascorbic acid was determined using the equation calculated from the calibration curve (y = 133.19x + 0.0064). Then, the percentage of ascorbic acid release (%) was determined using Equation (5).
(5)$$Ascorbic\;acid\;release\;\left(\%\right)=\frac{released\;ascorbic\;acid}{total\;load}\times 100$$

## 3. Results and Discussion

### 3.1. The Appearance of CS-Coated BC/HC Films

The appearance of BC film, BC/HC films, and BC/HC films coated with chitosan are illustrated in Figure 2. As seen, the virgin SCOBY BC film and CS film are opaque and yellowish transparent, respectively. From our previous XRD results [9,36], SCOBY BC exhibited higher crystallinity with a larger crystalline size than CS, confirming their opacity and transparent characteristics. In the presence of HC, BC/HC films exhibit transparency, indicating that HC, which is an amorphous polymer, is miscible with BC, hence modifying BC films physically (transparency) and mechanically (stiffness and elongation). After coating with chitosan, more yellowish and transparent films are obtained, resulting from the Mannich reaction between the acidic chitosan amino group and BC aldehyde group producing a yellowish Schiff base (imine) as shown in Figure 1. As a result of the Mannich reaction, chitosan is linked to BC via an imine (-C=N-) bond, creating a larger macromolecule. Moreover, chitosan coating is compatible with BC through intermolecular hydrogen bonding, as illustrated in Figure 1, making the CS/BC interaction even stronger. To prove that, a water absorption test was carried out. As shown in Figure 2, BC/HC films are completely swollen and deformed, indicating the absence of a physical cross-link between HC and BC due to the fact that HC, which is a low molecular weight molecule, lacks a number of hydroxyl groups to form intermolecular hydrogen bonding. In contrast, CS-coated BC/HC films exhibit dimension stability after the water absorption test, indicating that a CS/BC interpenetrating network (chemical crosslinker-free) is formed to prevent the penetration of water. The physically interpenetrating crosslinks between CS and BC are demonstrated in Figure 2. 

### 3.2. SEM Analysis

The cross-sectional SEM micrographs of BC, CS, and CS-coated BC/HC films are shown in Figure 3. As seen, cross-sectional SEM micrographs reveal that multilayers of BC sheets are observed, resulting from the layer-by-layer stacking of floating BC layers on the surface of the culture liquid. This unique multilayer architecture is not found in other BC cultures including *Nata de coco* (BC produced by *Komagataeibacter xylinus*). In the case of CS, a dense and continuous cross-section was observed. BC/HC films are found in a similar manner to BC film, indicating a homogenous BC/HC blend. In the case of the CS-coated BC90/HC10 film, a layer of CS is observed. Interestingly, in the case of CS-coated BC80/HC20 and CS-coated BC70/HC30 films, the CS layer is absent, indicating that CS is miscible with BC/HC films, resulting in CS/BC/HC blend films. 

### 3.3. ATR-FTIR Spectroscopy 

The FTIR spectrum (Figure 4a) of BC shows absorption bands at 3500–3125 cm^−1^, 3000–2900 cm^−1^, and 1650 cm^−1^ (weak) corresponding to the cellulose hydroxyl, cellulose C-H, and cellulose hemiacetal (reducing sugar) groups, respectively. The FTIR spectrum of the HC film (Figure 4b) shows absorption bands at 3500–3000 cm^−1^ (broader band, indicating fewer hydroxyl groups than BC), 3000–2900 cm^−1^ (CH_2_ vibration band), and 1650 cm^−1^ (strong band of collagen amide I group). The FTIR spectrum of the CS film (Figure 4c) shows a strong absorption band at 1560 cm^−1^ corresponding to the CS amino group. For CS-coated BC/HC films, the peak at 1650 cm^−1^ is assigned to-C=NH- (imine), judged by the strong yellowish color of coated films [40]. Therefore, the FTIR results provide supportive evidence, as shown in Figure 5 (mechanical properties), that changes in the physical and mechanical properties of CS-coated BC/HC films are due to the formation of an imine bond and the intermolecular H-bonding interaction among two types of compatible polymers. 

### 3.4. Mechanical Properties

Figure 5 shows the stress–strain curves of the BC films, CS films, BC/HC films, and CS-coated BC/HC films, and the mechanical properties are summarized in Table 1. The tensile strength values of the BC (film thickness: 53.00 ± 4.47 µm), CS (film thickness: 54.00 ± 5.48 µm), BC/HC films (film thickness: 42.50–46.25 µm), and CS-coated BC/HC films (film thickness: 108.75–120 µm) are 61.56 ± 4.71, 41.91 ± 0.97, (BC 90:HC10, 23.47 ± 1.23 MPa; BC 80:HC20, 18.02 ± 2.01 MPa; BC 70:HC30, 12.40 ± 0.98 MPa; and (BC 90:HC10: 1 wt% CS, 29.16 ± 1.90 MPa; BC 80:HC20: 1 wt% CS, 21.21 ± 1.59 MPa; BC 70:HC30: 1 wt% CS, 20.54 ± 3.47 MPa), respectively. From stress–strain curves, tensile strength tends to decrease with an increase in HC content. On the other hand, the percent elongation at break (%E) trend is opposite, implying that the BC/HC films exhibit a flexible property when compared to virgin BC film. After coating with 1 wt% CS, the films’ stiffness is slightly improved, resulting from the imine bonding and intermolecular H-bonding interaction. 

### 3.5. Antimicrobial Activity Evaluation

It is widely known that chitosan present in mild acidic conditions exists in a cationic form that exhibits antimicrobial activity against *S. aureus*. In this study, antimicrobial activity was evaluated according to the AATCC TM 147 standard method (disk diffusion method). The antimicrobial activity of the disk diffusion method was observed by a visible inhibition area, as shown in Figure 6. For representative BC film and representative BC/HC film, a colony of bacteria growth underneath the films is clearly visible, and a clear zone of inhibition is not observed. These indicate that both BC and HC are not active against *S. aureus*. When compared to CS-coated BC/HC films, it is found that CS-coated films exhibit a significant antimicrobial activity, confirmed by the absence of bacterial colonies beneath a representative testing film as well as the presence of an inhibited clear zone. The observed clear zone of inhibition is possibly due to the diffusion of acetic acid into agar media, which causes the mortality of *S. aureus*.

### 3.6. Ascorbic Acid Loading and Release

Ascorbic acid is an essential cofactor during collagen synthesis. It is a natural antioxidant but highly unstable when exposed (Figure 3). To maintain its active function, encapsulation is one approach, including wound dressing (Figure 6). However, loading capacity is always opposite to the release performance. As seen from Figure 7, CS adsorbs the highest amount of ascorbic acid (358 mg/g) due to the equilibrium shifting to the adsorbent (CS) driven by the opposite charge attraction (cationic chitosan and anionic ascorbic acid). In turn, the release performance of ascorbic acid against the attractive interaction between the adsorbent and the adsorbate (ascorbic acid) is too slow or even ineffective. Hence, CS is not appropriate for ascorbic acid delivery. For BC:HC blend films, ascorbic acid loading performance decreases in the following order: BC70:HC30 > BC80:HC20 > BC90:HC10. After chitosan coating, the loading performance is found in a similar manner, although it decreases further. As explained earlier, chitosan and BC form an interpenetrating network, preventing adsorbate penetration. In summary, an increase in the HC ratio results in an increase in ascorbic acid loading but at the cost of film deformation. Fortunately, this problem can be solved by chitosan coating. 

### 3.7. Ascorbic Acid Release Study

The percent cumulative release was measured by the summation of the percent concentration determined in each sampling. The percent cumulative release is plotted vs. sampling time as illustrated in Figure 8. As pointed out before, ascorbic acid is hardly released from the CS film when compared to its original uptake. Despite its high crystallinity, BC film performs well in ascorbic acid release due to its high water swellability. An improvement in water swellability was achieved by the addition of HC, leading to an increase in ascorbic acid mobility and releasing performance. It is noted that in the case of BC90:HC10, its corresponding values are lower than that of BC. The reason is that the released ascorbic acid was oxidized during the experiment (which is considered as human error) as confirmed by shifting in λ_max_@265 nm to the lower wavelength (Figure 8c). Hence, the loading/releasing values are relatively lower than those of BC. In opposite, the CS coating improves film dimensional stability but retards the diffusion of ascorbic acid. Another advantage of BC films including BC:HC blend films is that they can maintain active ascorbic acid for a long period of time.

## 4. Conclusions

SCOBY bacterial cellulose/hydrolyzed collagen (HC) films coated with chitosan were successfully prepared as described earlier. The resultant films were transparent, flexible, and resilient. FTIR spectra revealed that the BC hydroxyl band was broader in the presence of HC, arising from intermolecular hydrogen bonding between cellulose hydroxyl groups and collagen amide groups. The effect of HC on the physical and mechanical properties of BC was clearly explained. An improvement of the dimensional stability in a water environment was achieved by chitosan coating. The antibacterial activity test revealed that chitosan-coated BC/HC films exhibited excellent antimicrobial activity, representing a potential candidate for active wound dressing application. The results showed that BC:HC blend films exhibited ascorbic acid loading/releasing performance balance governed by the water swellability characteristic. Finally, another advantage of the BC films, including BC:HC blend films, was that they were able to maintain active ascorbic acid for a long period of time, probably due to the presence of plenty of BC hemiacetal reducing ends (protective property).

## Data Availability

Not applicable.

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
