# Peer review of "Chitosan-Coated Bacterial Cellulose (BC)/Hydrolyzed Collagen Films and Their Ascorbic Acid Loading/Releasing Performance: A Utilization of BC Waste from Kombucha Tea Fermentation"

_polymers, 2022, doi:10.3390/polym14214544_

Round 1

Reviewer 1 Report

In this manuscript, the BC: HC (hydrolyzed collagen) blend film coated with 1% chitosan (CS) was studied, and the release performance of the blend film was further studied by carrying ascorbic acid. The technology used in this study is very advanced, and the materials used are biodegradable. This study is very meaningful and provides candidate materials for studying active wound dressings. However, the author should address the following comments:

1.        In 1. Introduction,“However, BC is not considered as an active wound dressing due to its lack of antimicrobial activity and bioactive performance which is found in other bioactive materials such as chitosan/its derivatives, collagen, and etc. it is suggested that a detailed description of why hydrolyzed collagen was chosen to be blended with SCOBY BC to form the film and the rationale for choosing chitosan coating, and the current research progress related to this experiment.

2.        In 2.3. Preparation of chitosan coated BC/hydrolyzed collagen films, “The collected viscous paste was thinned by the addition of deionized water prior to casting onto the plastic mold (9" (L) x 8.5" (W)) and leaving to be dry freely.” Can you specify how much deionized water to add and the concentration of BC and HC?

3.        In 2.3. Preparation of chitosan coated BC/hydrolyzed collagen films, “Finally, the BC/HC film was coated with 1% (w/v) acidic chitosan solution and allowed to be dry freely in open air.” Can you specify the coating amount of 1% (w/ V) chitosan solution? Is the amount of chitosan solution applied to each group of films consistent?

4.        In 2.4. Characterizations, “Scanning electron microscopy (SEM) was performed using scanning electron microscope (SEM JEOL model JSM-6400LV, Japan) at an accelerating voltage of 15 kV.” No microscopic picture of the film is seen in the article.

5.        In 2.5.3 part, about the ascorbic acid loading efficiency, please check whether the formula “???????? ???? ??????? ?????????? (%) =” and

?????? ?? ????????e=” is accurate.

6.        In 2.5.3 part, in the formula “??????? ???????? =” , please check the spell of “sorbent”.

7.        In 2.5.3. Loading capacity of composite film and 2.5.4. Drug release tests, it is recommended to supplement the references of these two methods.

8.        In 2.5.3. Loading capacity of composite film, After that, the films were blotted with filter paper to remove excess water and measured by an UV-Vis spectrophotometer in the wavelength region of 200-600 nm. Could you explain what values were taken for the calculation at what wavenumber? Are duplicates done? Why is there no standard error?

9.        In 3.1. The appearance of CS coated BC/HC films, “The appearance of BC film, BC/HC films, and BC/HC films coated with chitosan are illustrated in Figure 1.” Please check whether the “Figure 1” representations are correct.

10.    In 3.1. The appearance of CS coated BC/HC films, “In the presence of HC, BC/HC films exhibit transparency, indicating that HC exhibited plasticizing effect on BC,” Is there a relationship between plasticizing effect and transparency? Is there some literature to support this idea?

11.    In 3.2. ATR-FTIR spectroscopy, “For CS coated BC/HC films, peak at 1650 cm-1 is assigned to be -C=NH- (imine) judged by the strong yellowish color of coated films.” It is suggested to add references there.

12.    In 3.2. ATR-FTIR spectroscopy, “Therefore, the FTIR results provide supportive evidence that changes in the physical and mechanical properties of CS coated BC/HC films are due to the formation of imine bond and the intermolecular H-bonding interaction among two types of compatible polymers.” This conclusion has not been preceded by an analysis of changes in mechanical properties, so it is not recommended to discuss mechanical properties here.

13.    In 3.3. Mechanical properties, it is suggested to reduce the mechanical description of data and increase the explanation of the reasons for the changes of indicators.

14.    In 3.4.  Antimicrobial activity evaluation, why is the antimicrobial activity evaluation only done for Gram-positive bacteria? It is recommended to add Gram-negative bacteria.

15.    In 3.6. Ascorbic acid release study, “An improvement in water swellability was achieved by the addition of HC, leading to an increase in ascorbic acid mobility and releasing performance.” The figure shows that BC90: HC10 films have lower ascorbic acid release properties than BC films and the reasons for this are not described and explained here.

16.    In 3.6. Ascorbic acid release study, can you explain the ascorbic acid release properties of doing CS in Figure 7a?

17.    In 3.6. Ascorbic acid release study, please check whether BC70: BC30, BC80: BC20, BC90: BC10 films in Figure 7 are correct.

18.    In 3.6. Ascorbic acid release study, it is recommended to change the color of the lines BC70: BC30:1%CS, BC80: BC20:1%CS, BC90: BC10 films:1%CS in Figure 7b, otherwise it will be confused with Figure 7a.

19.    In 4. Conclusions, “At first, BC: HC films by mechanical blending and casting onto a plastic mold. Finally, dried films were coated with 1%(w/v) acidic chitosan solution, resulting in transparent, flexible, and resilient films.” It is suggested not to describe the specific preparation method of the film in the conclusion.

20.     In 4. Conclusions, it is suggested to modify the logical cohesive words in the conclusion.

Author Response

In this manuscript, the BC: HC (hydrolyzed collagen) blend film coated with 1% chitosan (CS) was studied, and the release performance of the blend film was further studied by carrying ascorbic acid. The technology used in this study is very advanced, and the materials used are biodegradable. This study is very meaningful and provides candidate materials for studying active wound dressings. However, the author should address the following comments:

  1. In  Introduction,“However, BC is not considered as an active wound dressing due to its lack of antimicrobial activity and bioactive performance which is found in other bioactive materials such as chitosan/its derivatives, collagen, and etc.” it is suggested that a detailed description of why hydrolyzed collagen was chosen to be blended with SCOBY BC to form the film and the rationale for choosing chitosan coating, and the current research progress related to this experiment.

Ans : Reason for choosing chitosan and HC in this research is based on wound dressing application as highlighted in introduction part. In fact, the innovation of this work was to utilize SCOBY BC which is waste by-product from kombucha tea fermentation (expensive healthy drink as the main product)

  1. In 3.Preparation of chitosan coated BC/hydrolyzed collagen films, “The collected viscous paste was thinned by the addition of deionized water prior to casting onto the plastic mold (9" (L) x 8.5" (W)) and leaving to be dry freely.” Can you specify how much deionized water to add and the concentration of BC and HC?

Ans : the purpose of addition di water was to make the BC paste flowable and castable to obtain smooth and even film. The thickness is controlled by mold dimension. The more the amount of water the longer the drying time. So that we used the small amount of added water as we can to thin the paste.

  1. In 3.Preparation of chitosan coated BC/hydrolyzed collagen films, “Finally, the BC/HC film was coated with 1% (w/v) acidic chitosan solution and allowed to be dry freely in open air.” Can you specify the coating amount of 1% (w/ V) chitosan solution? Is the amount of chitosan solution applied to each group of films consistent?

Ans : 20 ml of 1% (w/ V) chitosan solution. The max conc. Of chitosan soln is 1 %w/v. Above this critical conc, chitosan solution will turn into gel which cannot be applied as a coating. Below this conc., BC substrate was swollen, resulting irregular shape of coated film. Therefore, the suitable conc. of 1 wt% was selected.

  1. In 4. Characterizations, “Scanning electron microscopy (SEM) was performed using scanning electron microscope (SEM JEOL model JSM-6400LV, Japan) at an accelerating voltage of 15 kV.”No microscopic picture of the film is seen in the article.

Ans: Thank you for pointing out. The SEM images are added in section 3.2.

  1. In 2.5.3 part, about the ascorbic acid loading efficiency, please check whether the formula “???????????? ??????? ?????????? (%) =” and

“?????? ?? ????????e=” is accurate.

Ans : yes

  1. In5.3part, in the formula “??????? ???????? =” , please check the spell of “sorbent”.

Ans : change to adsorbent, thank you.

  1. In 2.5.3. Loading capacity of composite filmand 2.5.4. Drug release tests, it is recommended to supplement the references of these two methods.

       Ans : ref. 39 is added

  1. In 2.5.3. Loading capacity of composite film, “After that, the films were blotted with filter paper to remove excess water and measured by an UV-Vis spectrophotometer in the wavelength region of 200-600 nm.”Could you explain what values were taken for the calculation at what wavenumber? Are duplicates done? Why is there no standard error?

Ans : Experimental was not clarified. We did re-write the part as highlighted. Since we report % accumulative release, graphs cannot be included SD. 

  1. In 3.1. The appearance of CS coated BC/HC films, “The appearance of BC film, BC/HC films, and BC/HC films coated with chitosan are illustrated in Figure 1.” Please check whether the “Figure 1” representations are correct.

Ans : Thank you. It is corrected.

  1. In 3.1. The appearance of CS coated BC/HC films, “In the presence of HC, BC/HC films exhibit transparency, indicating that HC exhibited plasticizing effect on BC,” Is there a relationship between plasticizing effect and transparency? Is there some literature to support this idea?

Ans : For polymer researchers, it is well-understood that plasticizing effect is a phenomenon caused by a small molecule such as dioctyl phthalate (DOP) additive. For example, during PVC compounding, DOP is added to plasticize (lubricate) PVC polymer chain to mobilize at molecular level (crystallinity change). Hence, we get transparent PVC product with flexibility (such as wrapping film..etc) . HC acts in a similar manner to DOP when adding to crystalline BC.  Not to get readers confused, we modify the writing as highlighted.

  1. In 3.2. ATR-FTIR spectroscopy, “For CS coated BC/HC films, peak at 1650 cm-1 is assigned to be -C=NH- (imine) judged by the strong yellowish color of coated films.” It is suggested to add references there.

Ans : ref 39

  1. In 3.2. ATR-FTIR spectroscopy, “Therefore, the FTIR results provide supportive evidence that changes in the physical and mechanical properties of CS coated BC/HC films are due to the formation of imine bond and the intermolecular H-bonding interaction among two types of compatible polymers.” This conclusion has not been preceded by an analysis of changes in mechanical properties, so it is not recommended to discuss mechanical properties here.

Ans : The first thought was that we just wanted to guide FTIR analysis to mechanical properties (Figure 5) as highlighted.

  1. In 3.3. Mechanical properties, it is suggested to reduce the mechanical description of data and increase the explanation of the reasons for the changes of indicators.

Ans : we are sorry that we don’t understand this comment.

  1. In 3.4.  Antimicrobial activity evaluation,why is the antimicrobial activity evaluation only done for Gram-positive bacteria? It is recommended to add Gram-negative bacteria.

Ans : Chitosan is a natural cationic polymer which is well-known for antimicrobial activity against gram positive bacteria such as S. aureus commonly found in infected skin wounds. Gram – is found in gastric system ( E. Coli). Moreover, chitosan is not good antibacterial for gram – due to its thick cell wall.

  1. In 3.6. Ascorbic acid release study, “An improvement in water swellability was achieved by the addition of HC, leading to an increase in ascorbic acid mobility and releasing performance.” The figure shows that BC90: HC10 films have lower ascorbic acid release properties than BC films and the reasons for this are not described and explained here.

Ans : Explanation is given as follows : It is noted that in case of BC90: HC10, its corresponding values are lower than that of BC. The reason is that the released ascorbic acid was oxidized as confirmed by shifting in lmax@265 nm to lower wavelength (Figure 8c). Hence, the loading/releasing values are relatively lower than those of BC.

  1. In 3.6. Ascorbic acid release study, can you explain the ascorbic acid release properties of doing CS in Figure 7a?

Ans : Explanation is highlighted in green in section 3.6.

  1. 17.In 6. Ascorbic acid release study, please check whether BC70: BC30, BC80: BC20, BC90: BC10 films in Figure 7 are correct.

Ans : corrected.

  1. In 3.6. Ascorbic acid release study, it is recommended to change the color of the lines BC70: BC30:1%CS, BC80: BC20:1%CS, BC90: BC10 films:1%CS in Figure 7b, otherwise it will be confused with Figure 7a.

Ans : we used the same color to compare between before and after CS coating.

  1. In 4. Conclusions, “At first, BC: HC films by mechanical blending and casting onto a plastic mold. Finally, dried films were coated with 1%(w/v) acidic chitosan solution, resulting in transparent, flexible, and resilient films.” It is suggested not to describe the specific preparation method of the film in the conclusion.

Ans : conclusion is written.

  1. In 4. Conclusions, it is suggested to modify the logical cohesive words in the conclusion.

Ans : modification made.

Reviewer 2 Report

The study presents the research results on ‘Characterizations of CS Coated BC/HC Films and Ascorbic Acid Loading/Releasing Performance’. In my view, the purpose of this manuscript should be emphasized more and this manuscript cannot be acceptable in its present form. The overall manuscript lacks in depth experimentation and lacks novelty. Please consider the following comments and suggestion for further revision.

1. A concise abstract is required. Also, more specific descriptions of authors' finding should be added in the abstract rather than overall result of study. The abstract should state briefly the purpose of the research, the principal results and major conclusions. 

2. There are many research results about the CS Coated BC with various chemicals. What is the main advantages of this research? Please provide the advantage of this film and compare to other recent research papers. Please provide the novelty of this study and compare to other recent research papers in the view of detection limit in a table. 

3. Please explain in detail why you applied HC and CS to BC. 

4, It is necessary to add content related to strain and culture condition for BC production. It is necessary to explain the productivity of BC compared to other studies, and add an explanation for the difference. 

5. It is necessary to explain in detail why the cumulative release increases at 6 hours in Fig. 7. It is necessary to modify the colors of each figure in Fig. 7(c) to make it easier to distinguish them. It is necessary to explain in detail the meaning of the difference in absorbance values according to BC70/BC80/BC90, and it is also necessary to add an explanation as to whether the difference is significant.

6. The explanation of results and discussion are not enough. Overall, this paper lacks the actual discussion of experimental data. They simply described their data. A more in-depth discussion other than description is needed to improve for each section of results by making a comparison with other studies to elucidate the merit of the developed bacteria cellulose.

7. Authors should discuss the potential industrial application of this technology in an economical and technical viewpoint. 

8. In order to improve readers' understanding and emphasize the originality of the results of this study, it is necessary to describe in detail the future prospects that can overcome the uncertainty of this technology, including the background and reasons in the ‘Results & Discussion’ part and the ‘Conclusion’ part 

9. There are many figures and tables. Only major results should be included in the MS and the others should be described in supplementary data file.

Author Response

The study presents the research results on ‘Characterizations of CS Coated BC/HC Films and Ascorbic Acid Loading/Releasing Performance’. In my view, the purpose of this manuscript should be emphasized more and this manuscript cannot be acceptable in its present form. The overall manuscript lacks in depth experimentation and lacks novelty. Please consider the following comments and suggestion for further revision.

 Ans : we change the title of paper to “Chitosan coated bacterial cellulose (BC)/hydrolyzed collagen films and their ascorbic acid loading/releasing performance: A utilization of BC waste from kombucha tea fermentation”. The novelty of this work is the utilization of BC waste from kombucha tea fermentation to new, high value-added, and green waste. No report concerning the usage of SCOBY BC which is the waste when compared Nato de coco which is food (main product).

  1. A concise abstract is required. Also, more specific descriptions of authors' finding should be added in the abstract rather than overall result of study. The abstract should state briefly the purpose of the research, the principal results and major conclusions. 

Ans : 200 words abstract is very short and conclusive, in my view.  Major findings included achievement of BC with excellent properties (encapsulation ability of highly sensitive ascorbic acid, excellent mechanical properties) from SCOBY BC waste.

  1. There are m words any research results about the CS Coated BC with various chemicals. What is the main advantages of this research? Please provide the advantage of this film and compare to other recent research papers. Please provide the novelty of this study and compare to other recent research papers in the view of detection limit in a table. 

Ans : as comprehensively explained in the revised version.

  1. Please explain in detail why you applied HC and CS to BC. 

Ans : as comprehensively explained in the revised version

4, It is necessary to add content related to strain and culture condition for BC production. It is necessary to explain the productivity of BC compared to other studies, and add an explanation for the difference. 

Ans : as explained in the reversion. SCOBY BC is a waste or by-product. The main product is kombucha tea. In this research, we prepared kombucha tea and get BC as waste for our research. You may miss the main point.

  1. It is necessary to explain in detail why the cumulative release increases at 6 hours in Fig. 7. It is necessary to modify the colors of each figure in Fig. 7(c) to make it easier to distinguish them. It is necessary to explain in detail the meaning of the difference in absorbance values according to BC70/BC80/BC90, and it is also necessary to add an explanation as to whether the difference is significant.

Ans : as explained to referee 1.

  1. The explanation of results and discussion are not enough. Overall, this paper lacks the actual discussion of experimental data. They simply described their data. A more in-depth discussion other than description is needed to improve for each section of results by making a comparison with other studies to elucidate the merit of the developed bacteria cellulose.

Ans : SCOBY BC is waste. Our purpose is not to produce BC. But to use it as raw material (green waste).

  1. Authors should discuss the potential industrial application of this technology in an economical and technical viewpoint..

Ans : another next step.

  1. In order to improve readers' understanding and emphasize the originality of the results of this study, it is necessary to describe in detail the future prospects that can overcome the uncertainty of this technology, including the background and reasons in the ‘Results & Discussion’ part and the ‘Conclusion’ part 

Ans : all comments are focused on the preparation of SCOBY BC as a product. However, it is a by-product. Pls study the kombucha tea fermentation.

  1. There are many figures and tables. Only major results should be included in the MS and the others should be described in supplementary data file.

Ans : Polymers is OPAC and paid for publication. Polymers has no such limitation of figures, pages, and words. We would like achieve high viewing traffic.

Reviewer 3 Report

The manuscript entitled “Characterizations of CS Coated BC/HC Films and Ascorbic Acid Loading/Releasing Performance" study BC : HC (hydrolyzed collagen) blend films coated with 1 wt% chitosan (CS) loaded with ascorbic acid to study loading / releasing performance. However, the presented article could be published after major revision for the following.

1.      The abbreviations in the title should be spelled out.

2.      Introduction part should be enriched with a detailed background as well as a justification for the current research.

3.      Section 2.5.4. UV-Vis measurement spectrophotometer instrument company should be mentioned. The wavelength should be mentioned instead of using wavelength range.

4.      Scheme 3: Correct the structure of L-dehydroascorbic acid.

5.      Additional techniques for film characterization such as XRD should be used besides ATR-FTIR spectroscopy.

6.      Discussion part is weak and needs to be improved with details supported by recent references.

7.      Finally, English should be polished throughout the text

Author Response

Comments and Suggestions for Authors

The manuscript entitled “Characterizations of CS Coated BC/HC Films and Ascorbic Acid Loading/Releasing Performance" study BC : HC (hydrolyzed collagen) blend films coated with 1 wt% chitosan (CS) loaded with ascorbic acid to study loading / releasing performance. However, the presented article could be published after major revision for the following.

  1. The abbreviations in the title should be spelled out.

Ans : Thank you for comment.

  1. Introduction part should be enriched with a detailed background as well as a justification for the current research.

               Ans : Introduction is re-written to be fit with new title. We provide information about SCOBY BC as a by-product. So that it is a green waste as raw material in this study.

  1. Section 2.5.4. UV-Vis measurement spectrophotometer instrument company should be mentioned. The wavelength should be mentioned instead of using wavelength range.

ANS: Thank you. It is done.

  1. Scheme 3: Correct the structure of L-dehydroascorbic acid.

ANS: Thank you. It is done.

  1. Additional techniques for film characterization such as XRD should be used besides ATR-FTIR spectroscopy.

ANS: we referred to our previous publications (9, 35).

  1. Discussion part is weak and needs to be improved with details supported by recent references.

Ans : rewritten is highlighlighted.

  1. Finally, English should be polished throughout the text

Ans : Thank you

Round 2

Reviewer 2 Report

The issues raised by review have been well addressed, and I recommend its publication in the journal.

Reviewer 3 Report

The article could be accepted in its present form.